# Tiludronate concentrations and cytologic findings in synovial fluid after intravenous regional limb perfusion with tiludronate in horses

Barbara G. Hunter*, Katja F. Duesterdieck-Zellmer and
Maureen K. Larson

Department of Clinical Sciences, College of Veterinary Medicine, Oregon State University,
Corvallis, OR, USA
* Current affiliation: Matamata Veterinary Services Equine, Matamata, Waikato, New Zealand

Corresponding author
Katja F. Duesterdieck-Zellmer,
katja.zellmer@oregonstate.edu

## ABSTRACT

Anecdotal accounts of tiludronate administration via intravenous regional limb perfusion (IVRLP) exist despite a lack of information regarding safety for synovial structures in the perfused area. The objective of this study was to determine whether tiludronate concentrations in synovial structures after IVRLP with low dose (0.5 mg, LDT) or high dose (50 mg, HDT) tiludronate remain below a value demonstrated *in vitro* to be safe for articular cartilage (<19,000 ng/ml), and to determine effects of tiludronate on synovial fluid cytology variables compared to saline perfused control limbs. Using a randomized controlled experimental study design, horses received IVRLP with LDT ($n = 6$) or HDT ($n = 6$) in one forelimb and IVRLP with saline in the contralateral limb. Synovial fluid cytology variables and tiludronate concentrations were evaluated in navicular bursae (NB), and distal interphalangeal (DIP) and metacarpophalangeal (MCP) joints one week before and 30–45 min after IVRLP, and in DIP and MCP joints 24 h after IVRLP. Data were analyzed with 2-way rmANOVA ($p < 0.05$). Highest measured synovial fluid tiludronate concentrations occurred 30–45 min post-perfusion. Mean tiludronate concentrations were lower in LDT limbs (MCP = 39.6 ± 14.3 ng/ml, DIP = 118.1 ± 66.6 ng/ml, NB = 82.1 ± 30.2 ng/ml) than in HDT limbs (MCP = 3,745.1 ± 1,536.6 ng/ml, DIP = 16,274.0 ± 5,460.2 ng/ml, NB = 6,049.3 ± 1,931.7 ng/ml). Tiludronate concentration was >19,000 ng/ml in DIP joints of two HDT limbs. Tiludronate was measurable only in synovial fluid from HDT limbs 24 h post-perfusion. There were no differences in synovial fluid cytology variables between control and treated limbs. **Conclusions.** In some horses, IVRLP with HDT may result in synovial fluid concentrations of tiludronate that may have adverse effects on articular cartilage, based on *in vitro* data. IVRLP with LDT is unlikely to promote articular cartilage degradation. Further studies to determine a safe and effective dose for IVRLP with tiludronate are needed.

## INTRODUCTION

Tiludronate is a non-nitrogenous bisphosphonate used in humans for the treatment of Paget's disease and osteoporosis, because it normalizes bone-turnover at therapeutic doses (*Bonjour et al., 1995*). In recent years, tiludronate has been utilized in horses for treatment of diseases related to abnormal bone remodeling (*Kamm, McIlwraith & Kawcak, 2008*) and has reportedly been effective in some horses in reducing pain associated with navicular disease (*Denoix, Thibaud & Riccio, 2003*), distal hock osteoarthritis (*Gough, Thibaud & Smith, 2010*) and thoracolumbar osteoarthritis (*Coudry et al., 2007*). Complications associated with systemic administration of tiludronate include mild tachycardia during administration and transient hypocalcemia following injection (*Varela et al., 2002*). Anecdotally, signs of colic and acute renal failure have also been encountered. Presumably to decrease the occurrence of complications associated with systemic administration of tiludronate, and to decrease the cost of treatment, equine practitioners are currently administering tiludronate via intravenous regional limb perfusion (IVRLP) for the treatment of navicular disease and other orthopaedic conditions of the distal limb (*Carpenter, 2012*). Doses that are anecdotally being used for IVRLP however, such as 50 mg per perfusion, are lacking evaluation of safety for tissues within the perfused region.

Although the target organ for bisphosphonates is bone, tiludronate also has effects on articular cartilage that are concentration dependent. Concentrations of $\geq$ 19,000 ng/ml enhanced chondrocyte apoptosis and proteoglycan release in equine articular cartilage explants (*Duesterdieck-Zellmer, Driscoll & Ott, 2012*). Administration of medications such as antibiotics via intravenous regional limb perfusion allows veterinarians to attain considerably higher tissue and synovial fluid concentrations than can be achieved with systemic administration (*Rubio-Martinez & Cruz, 2006*). Assuming that IVRLP with tiludronate follows similar pharmacokinetics, cartilage within the perfused area may potentially be exposed to synovial fluid tiludronate concentrations that promote cartilage degradation. Therefore, the objectives of the current study were to determine tiludronate concentrations achieved in distal limb synovial structures following intravenous regional limb perfusion with a low (0.5 mg; LDT) or high (50 mg; HDT) dose of tiludronate and to determine the effects of these two dosing regimens on synovial fluid cytology variables compared to placebo controlled limbs. The working hypothesis was that IVRLP with LDT would result in synovial fluid concentrations that were safe for cartilage in vitro ($<$19,000 ng/ml; *Duesterdieck-Zellmer, Driscoll & Ott, 2012*) and that synovial fluid cytology variables would not vary significantly from controls. It was further hypothesized that IVRLP with 50 mg of tiludronate (HDT), the lowest of three doses currently used in clinical practice to the knowledge of these authors, would result in synovial fluid concentrations that are unsafe for cartilage in vitro ($\geq$ 19,000 ng/ml; *Duesterdieck-Zellmer, Driscoll & Ott, 2012*) and that synovial fluid cytology variables in treated limbs would differ from control limbs.

## MATERIALS AND METHODS

### Animals

All experimental procedures were performed with the approval of the Institutional Animal Care and Use Committee of Oregon State University (ACUP# 4280 and 4459). Six adult healthy horses (mean body weight, 600 kg; range, 514–636 kg; mean age, 10.6 years; range 4–17 years; breed, 1 Thoroughbred, 3 Quarter Horses, 1 Warmblood, 1 Quarter Horse cross) were used for the LDT trial and one year later, six different healthy horses (mean weight, 480 kg, range, 414–545 kg; mean age, 12.5 years; range, 12–19 years; breed, 1 Rocky Mountain Horse, 1 Appaloosa, 1 Morgan, 1 Quarter Horse, 1 American Paint Horse and 1 Arabian cross) were used for the HDT trial. All horses were graded for lameness from 0 to 5 according to guidelines by the American Association of Equine Practitioners (*Stashak, 2002*). Further, response to lower front limb flexion tests was recorded. All horses were sound at a walk, but most showed mild front limb lameness at the trot. Horses were housed in box stalls or small paddocks, given free access to grass hay and water and underwent daily physical exams for the duration of the experiments. None of the horses had ever received tiludronate prior to these experiments. Systematic assessment of initial comparability between treated and control limbs with respect to lameness or synovial cytologic response variables was not performed.

### Experimental protocol

After initial examination, horses were sedated with detomidine (Dormosedan; Pfizer Animal Health, New York, New York, USA; 0.01–0.015 mg/kg IV) and were given additional doses of detomidine and/or butorphanol (Torbugesic; Zoetis Animal Health, Florham Park, New Jersey, USA; 0.01 mg/kg) as needed. Distal forelimbs were locally anesthetized using high four-point perineural blocks with bupivacaine (Marcaine; Hospira Inc, Lake Forest, Illinois, USA) to facilitate sample collection and IVRLP. Baseline synovial fluid samples were collected aseptically from both distal interphalangeal and metacarpophalangeal joints and from the navicular bursae under radiographic guidance (*Boyce et al., 2010*). Distal limbs were bandaged for 24 h following synoviocenteses.

Seven days later, horses were sedated and distal forelimbs were locally anesthetized as described above. IVRLP was performed aseptically by an investigator who was blinded to treatment allocations (BGH) on one randomly assigned front limb either with 0.5 mg tiludronate (Tildren; CEVA, Libourne, France; $n = 6$; LDT) or with 50 mg tiludronate ($n = 6$; HDT), both diluted in 50 ml saline. The contralateral forelimb received IVRLP with 50 ml saline alone to serve as a placebo control. Briefly, rolled gauze pads were placed on either side of the flexor tendons in the proximal half of the metacarpus, followed by application of a 10.2-cm wide rubber tourniquet (Esmark Bandage; Cardinal Health, McGraw Park, Illinois, USA) to cover 15–18 cm of the metacarpus. Tourniquets were applied as tightly as possible by the same investigator (BGH) each time. A 21-gauge 1.9-cm butterfly catheter (Surflo Winged Infusion Set; Terumo Corporation, Tokyo, Japan) was inserted into a palmar digital vein and the perfusate was injected over 3–5 min. Catheters were removed immediately following infusions and a temporary bandage of gauze and

elastic wrap (Coflex, Andover, Salisbury, Massachusetts, USA) was applied over the venipuncture site. Bandages and tourniquets were left in place for 30 min.

Just prior to tourniquet removal, 10 ml of venous blood was obtained from the jugular vein for analysis of serum tiludronate concentrations. Immediately following tourniquet removal, synovial fluid samples were obtained from distal interphalangeal and metacarpophalangeal joints, as well as the navicular bursae as described above. Synoviocentesis of distal interphalangeal and metacarpophalangeal joints was performed simultaneously by 2 investigators (BGH and KFDZ) at 30–35 min and synoviocentesis of the navicular bursa was performed at 35–45 min after injection of the perfusate.Distal limbs were subsequently bandaged.

Twenty-four hours following IVRLP, horses were evaluated for lameness as described above by an investigator blinded to treatment allocation (BGH). Horses treated with LDT were sedated again and synovial fluid was collected as described from both distal interphalangeal and metacarpophalangeal joints. Subsequently, daily lameness examinations were performed for an additional seven days before being released from the study. In horses treated with HDT, synovial fluid was collected from the same joints, after euthanasia via intravenous injection of pentobarbital (Beuthanasia-D Special; Schering Plough Animal Health, Kenilworth, New Jersey, USA; 87 mg/kg IV).

## Sample processing and analysis

All sample processing and analyses were performed by personnel blinded to treatment allocation. Following sample collection, 200–300 µl of synovial fluid was placed in a 2 ml vial containing 7.5% EDTA liquid (Monoject; Tyco Healthcare, Mansfield, Massachusetts, USA) for cytology analysis. Total solids were measured using a refractometer (E-line Veterinary, Bellingham+Stanley, Basingstoke, Hampshire, UK). Total nucleated cell counts were determined manually (BMP Leuko-Tik, BMP Biomedical Polymers, Gardner, Massachusetts, USA). Differential cell counts were performed on Wright Giemsa stained cytospin slides (CytoSpin* 4 Cytocentrifuge; Thermo Scientific, Waltham, Massachusetts, USA).

Jugular blood samples were allowed to clot at room temperature for 30 min and centrifuged at 3,500x$g$ for 5 min. Serum was separated and frozen at −80 °C until tiludronate analysis. Synovial fluid was centrifuged at 10,000x$g$ for 30 min at 4 °C, the supernatant was aspirated and frozen at −80 °C until tiludronate analysis.

Tiludronate analysis was performed as previously described (*Duesterdieck-Zellmer et al., 2014*) using high performance liquid chromatography (XBridge phenyl column, Waters, Milford, Massachusetts, USA) followed by mass spectrometry (API 4000; Applied Biosystems, Grand Island, New York, USA) . Briefly, tiludronate in all samples was methylated with 0.2 M trimethylsilyldiazomethane in acetone. Concentrations of methylated tiludronate were determined against known standard samples (10–64 ng/ml). An additional standard sample with a tiludronate concentration of 0.5 ng/ml was used to determine presence or absence of tiludronate below the linear part of the standard curve. For synovial fluid samples, tiludronate standards were procured in equine synovial fluid

and for serum samples standards were generated in equine serum from untreated horses euthanized for reasons unrelated to this study. The same standards were used for low dose trial samples as for high dose trial samples. Serial dilutions were performed on all samples with tiludronate concentrations above the range of the standard curve. All unknown and standard curve samples were spiked with a known amount of deuterated tiludronate (Toronto Research Chemicals, Toronto, Ontario, Canada) as an internal control. Further, positive and negative control samples were run concurrently with each batch of samples. The lower limit of accurate quantification of the assay was 10 ng/ml.

### Data analysis

Results for cytology variables and lameness grades are reported as mean ± standard error and for tiludronate concentrations as mean [lowest-highest measurement]. Normal distribution of data was assured using the Anderson-Darling normality test. Cytology variables for each joint and lameness grades for each limb were compared to baseline values over time and between treated and control limbs for the low dose trial and the high dose trial separately using two-way repeated measures ANOVA followed by Holm-Sidak's multiple comparisons tests. Statistical significance was set at $P \leq 0.05$ and analyses were performed in Graphpad Prism (Graph Pad Software, San Diego, California, USA).

## RESULTS

All horses were sound at a walk prior to treatment and remained sound at a walk after treatment. Mean lameness grade seven days prior to treatment was 0.8 ± 1.0 and 0 ± 0 for LDT and control limbs, respectively and 0.5 ± 0.3 and 0 ± 0 for HDT and control limbs, respectively. No differences over time or between treated and control limbs were found in horses treated with LDT. Limbs treated with HDT had significantly higher lameness scores the day after IVRLP (2.0 ± 0.5) compared to seven days prior to IVRLP ($p = 0.0048$) and this was not the case for the control limbs. Further, limbs treated with HDT had significantly higher lameness scores than control limbs (0.3 ± 0.3) on the day after IVRLP ($p = 0.0036$). In one horse with a limb that was positive to distal limb flexion prior to IVRLP with LDT, the positive response to distal limb flexion was unchanged after IVRLP. No other horses displayed positive distal limb flexion tests before or after IVRLP.

Synovial fluid total solids concentration (Fig. 1) was not significantly different in any synovial structure between LDT and control limbs, or between HDT and control limbs, and there was no difference over time in total solids concentration in any synovial structure of treated or control limbs.

Total nucleated cell count in synovial fluid (Fig. 2) was not significantly different in any synovial structure between LDT and control limbs, or between HDT and control limbs. However, 24 h after IVRLP, total nucleated cell count was increased in the metacarpophalangeal joints of limbs treated with LDT, HDT, and saline. In the distal interphalangeal joint, total nucleated cell count was increased in saline treated limbs of the HDT group.

Synovial fluid neutrophil count (Table 1) was not significantly different in any synovial structure between LDT and control limbs, or between HDT and control limbs, and there

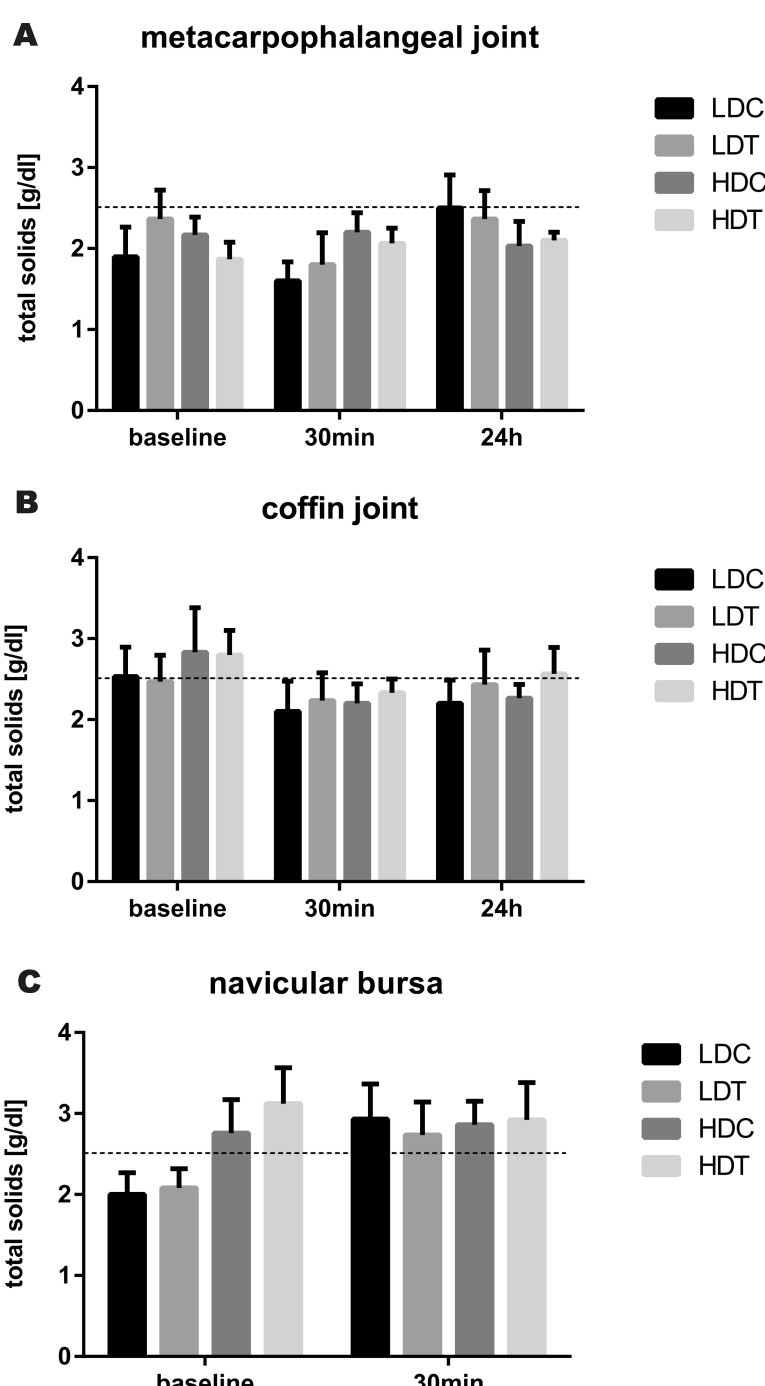

**Figure 1** **Bar graphs illustrating synovial fluid total solids concentrations after IVRLP with tiludronate or saline.** Synovial fluid total solids concentrations over time after IVRLP with 0.5 mg (LDT) or 50 mg (HDT) tiludronate diluted in 50 ml saline in one randomly assigned forelimb or with 50 ml saline in the contralateral forelimb as control for the low dose (LDC) or as control for the high dose of tiludronate (HDC). Synovial fluid was sampled from the metacarpophalangeal joint (A), the coffin joint (B) and the navicular bursa (C). The dotted line represents the upper limit of the normal reference interval (*Davidson & Orsini, 2007*). Error bars represent SEM.

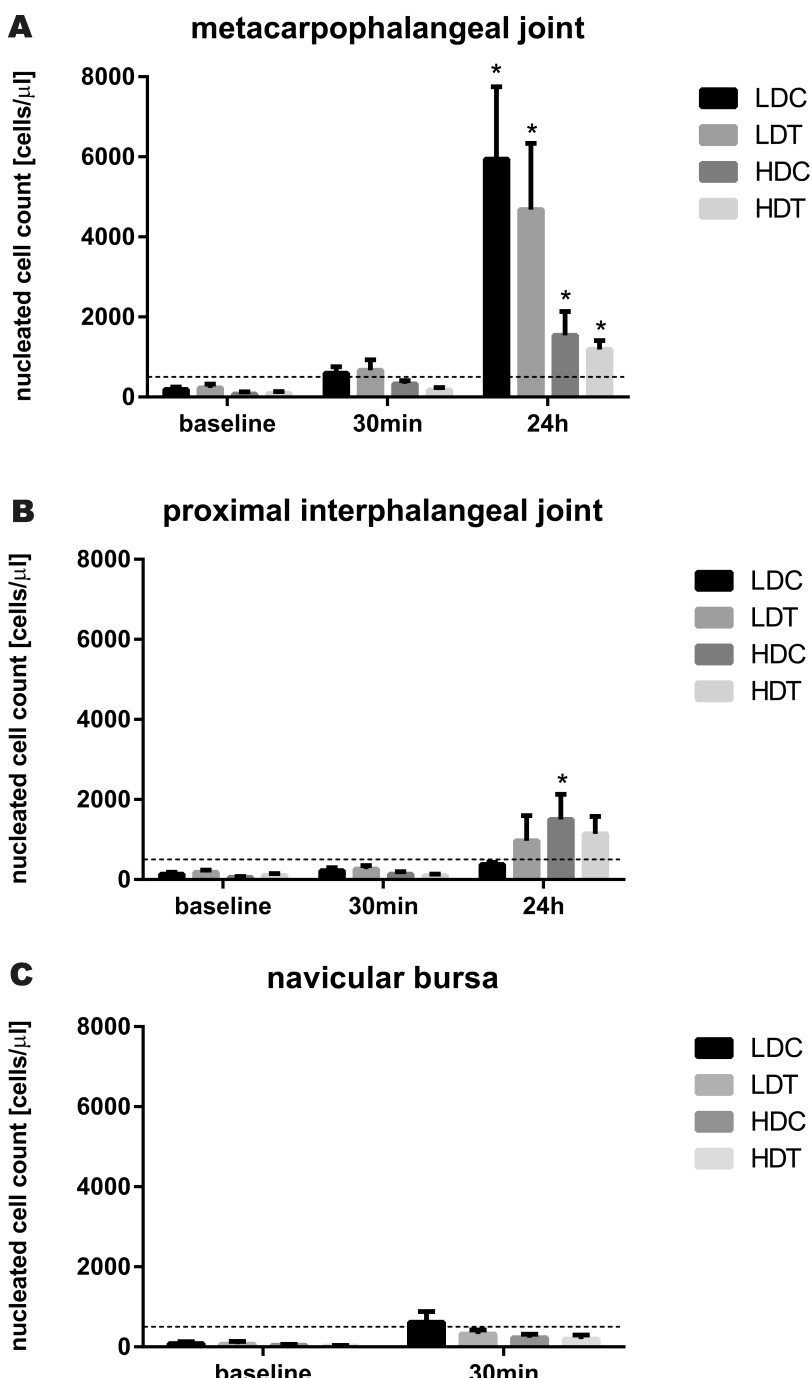

**Figure 2** **Bar graphs illustrating synovial fluid total nucleated cell counts after IVRLP with tiludronate or saline.** Synovial fluid total nucleated cell counts over time after IVRLP with 0.5 mg (LDT) or 50 mg (HDT) tiludronate diluted in 50 ml saline in one randomly assigned forelimb or with 50 ml saline in the contralateral forelimb as control for the low dose (LDC) or as control for the high dose of tiludronate (HDC). Synovial fluid was sampled from the metacarpophalangeal joint (A), the coffin joint (B) and the navicular bursa (C). The dotted line represents the upper limit of the normal reference interval (*Mahaffey, 2002*). An asterisk indicates significant difference ($P < 0.05$) from baseline measurement. Error bars represent SEM.

Table 1  **Table of percentage of neutrophils in synovial fluid.** Mean percentage and standard error of the mean of neutrophils among all nucleated cells in synovial fluid over time after IVRLP with 0.5 mg (LDT) or 50 mg (HDT) tiludronate diluted in 50 ml saline in one randomly assigned forelimb or with 50 ml saline in the contralateral forelimb as control for the low dose (LDC) or as control for the high dose of tiludronate (HDC). *Mahaffey (2002)* suggests that the proportion of neutrophils in normal synovial fluid should not exceed 10%, except in samples with very low cell counts.

| | Metacarpophalangeal joint | | |
| --- | --- | --- | --- |
| | Baseline | 30–35 min | 24 h |
| LDC | 2.3 [1.1] | 3.3 [2.0] | 16.8 [5.9] |
| LDT | 9.2 [9.0] | 6.5 [4.2] | 22.8 [6.2] |
| HDC | 5.3 [1.8] | 5.3 [1.8] | 5.3 [1.8] |
| HDT | 5.8 [3.6] | 0.7 [0.4] | 5.8 [2.1] |

| | Distal interphalangeal joint | | |
| --- | --- | --- | --- |
| | Baseline | 30–35 min | 24 h |
| LDC | 18.7 [6.9] | 6.5 [1.9] | 10.8 [4.9] |
| LDT | 22.0 [10.1] | 22.3 [7.6] | 8.8 [2.5] |
| HDC | 1.3 [0.9] | 10.5 [5.1] | 13.0 [3.1] |
| HDT | 5.5 [3.2] | 16.2 [9.7] | 6.8 [2.1] |

| | Navicular bursa | |
| --- | --- | --- |
| | Baseline | 35–45 min |
| LDC | 8.7 [3.8] | 13.7 [8.0] |
| LDT | 11.3 [4.3] | 8.1 [5.4] |
| HDC | 16.6 [10.3] | 25.8 [14.6] |
| HDT | 9.0 [7.2] | 13.2 [8.0] |

was no difference over time in neutrophil count in any of the synovial structures of treated or control limbs.

Tiludronate concentrations in synovial fluid are reported in Table 2. Tiludronate was not detected in any samples prior to IVRLP. In horses receiving LDT, tiludronate was detectable in synovial fluid of saline perfused limbs immediately after tourniquet release, albeit at concentrations below the lower limit of accurate quantification of the assay. Tiludronate concentration was detected at higher concentrations in synovial fluid of the LDT limbs immediately after tourniquet release, but was no longer detectable by 24 h post-perfusion. Highest tiludronate concentrations were measured in synovial fluid samples from HDT limbs immediately after tourniquet release, and tiludronate was still present in samples taken 24 h after IVRLP. In control limbs of HDT horses, tiludronate was present in synovial fluid immediately after tourniquet release. However, 24 h after IVRLP, tiludronate was detected only below the lower limit of accurate quantification of the assay in saline treated limbs of horses receiving HDT.

In HDT horses, tiludronate concentration in serum immediately prior to tourniquet release was 111.5 ng/ml [0.0–300.0] and in LDT horses, serum tiludronate concentration was below the lower limit of accurate quantification of the assay.

**Table 2 Table of synovial fluid tiludronate concentrations.** Mean [lowest-highest measurement] tiludronate concentrations in ng/ml in synovial fluid before and after IVRLP with tiludronate or saline. An asterisk indicates that tiludronate was detectable in synovial fluid, albeit below the level of quantitation for the assay (10 ng/ml).

| | Metacarpophalangeal joint | | |
|---|---|---|---|
| | Baseline | 30–35 min | 24 h |
| LDC | 0.0 | 0.0* | 0.0 |
| LDT | 0.0 | 39.6 [0.0–101.8] | 0.0 |
| HDC | 0.0 | 24.6 [0.0–67.3] | 0.0* |
| HDT | 0.0 | 3,745.1 [763.2–10,850.0] | 70.8 [48.4–92.7] |

| | Distal interphalangeal joint | | |
|---|---|---|---|
| | Baseline | 30–35 min | 24 h |
| LDC | 0.0 | 0.0* | 0.0 |
| LDT | 0.0 | 118.1 [26.6–449.6] | 0.0 |
| HDC | 0.0 | 76.6 [38.9–155.6] | 0.0* |
| HDT | 0.0 | 16,274.0 [4,390.0–33,700.0] | 16.5 [0.0–30.6] |

| | Navicular bursa | |
|---|---|---|
| | Baseline | 35–45 min |
| LDC | 0.0 | 0.0* |
| LDT | 0.0 | 82.1 [0.0–195.8] |
| HDC | 0.0 | 89.0 [12.6–182] |
| HDT | 0.0 | 6,049.3 [211.0–11,187.0] |

## DISCUSSION AND CONCLUSIONS

Before IVRLP with tiludronate should be investigated further as a possible therapy for distal limb orthopedic disease in horses, it should first be determined whether tiludronate concentrations achieved in the perfused area are safe for the perfused tissues. This study determined synovial fluid concentrations of tiludronate after IVRLP with 0.5 or 50 mg tiludronate diluted in 50 ml saline as a first step in assessing the safety of this technique for articular cartilage in the perfused area. A previously published in vitro concentration–response study suggested that synovial fluid tiludronate concentrations of ≥19,000 ng/ml may be unsafe for healthy articular cartilage and concentrations of ≥1,900,000 ng/ml may be unsafe for osteoarthritic cartilage (*Duesterdieck-Zellmer, Driscoll & Ott, 2012*). While mean synovial fluid tiludronate concentrations in the present study did not exceed 19,000 ng/ml, recorded concentrations varied greatly between individual horses and synovial structures. Synovial fluid tiludronate concentration was >30,000 ng/ml upon tourniquet release in the distal interphalangeal joint of two limbs treated with HDT. This is concerning since concentrations of this magnitude increased chondrocyte apoptosis and proteoglycan release from articular cartilage matrix *in vitro* (*Duesterdieck-Zellmer, Driscoll & Ott, 2012*). However, these findings have not yet been

confirmed in vivo. Synovial fluid tiludronate concentrations >2,000,000 ng/ml after intraarticular administration of 50 mg tiludronate into equine middle carpal joints were not associated with worse cartilage histology scores or decreased proteoglycan content at two weeks after treatment when compared to saline treated joints in four horses (*Duesterdieck-Zellmer et al., 2014*). Nevertheless, tiludronate treated joints showed temporarily increased proteoglycan degradation and amelioration of increases in aggrecan synthesis compared to control joints, as determined by synovial fluid biomarkers. Although the clinical significance of these findings was uncertain, it is possible that IVRLP with 50 mg of tiludronate diluted to 50 ml with saline may not be safe for articular cartilage in the distal interphalangeal joints of some horses.

Great variability in synovial fluid drug concentrations after IVRLP has been previously reported (*Murphey, Santschi & Papich, 1999*; *Butt et al., 2001*; *Parra-Sanchez et al., 2006*; *Levine et al., 2010*; *Hyde et al., 2013*; *Mahne et al., 2014*) and has been attributed to leakage of perfusate from the distal limb across the tourniquet, which occurred to a small extent in all horses of this study, and variable drug doses on a per bodyweight basis (*Butt et al., 2001*). Perivascular injection of perfusate could also contribute to variability in synovial fluid tiludronate concentrations, but in the present study, injection of perfusate was interrupted at the first sign of this occurring. Subsequently, the coaxial palmar digital vein of the same limb was used to complete the injection, while an assistant was holding off the initial venipuncture site. Interestingly, horses with the lowest serum concentrations of tiludronate prior to tourniquet release were not consistently found to have the highest synovial fluid tiludronate concentrations in treated limbs, suggesting that serum drug concentration is only a marginal indicator of synovial fluid tiludronate concentration after IVRLP.

Highest tiludronate concentrations were measured in synovial fluid from distal interphalangeal joints, followed by the navicular bursa and finally the metacarpophalangeal joints of HDT and LDT limbs. A similar pattern has been described for synovial fluid concentrations of antibiotics after IVRLP (*Butt et al., 2001*; *Rubio-Martinez et al., 2006*). Possible explanations for this phenomenon include the injection of perfusate at the level of the metacarpophalangeal joint in a distad direction, favoring perfusion of more distally located synovial structures, as well as differences in ratio of synovial fluid volume to synovial lining surface (*Rubio-Martinez et al., 2006*).

As expected, synovial fluid tiludronate concentrations increased in a dose dependent fashion in the present study. Highest concentrations were found immediately following tourniquet removal and negligible concentrations were documented in synovial fluid 24 h after IVRLP. In a relevant report, radioactively labeled bisphosphonate (technetium Tc 99m medronate) followed a similar pattern after intraarticular administration into equine antebrachiocarpal joints, and elimination of the bisphosphonate from the joint space was suggested to occur via transfer from synovial fluid to plasma (*Dulin et al., 2012*). Although this has not been assessed, it is conceivable that bisphosphonates may also diffuse into articular cartilage and subchondral bone, following their affinity for calcium. Detection of tiludronate at low concentrations in control joints as was documented in the present study has also been reported after intraarticular injection of tiludronate in horses

(*Duesterdieck-Zellmer et al., 2014*), and likely reflected a mild degree of tourniquet escape during IVRLP, as well as redistribution from perfused tissues into systemic circulation after tourniquet release. In comparison, serum concentrations prior to tourniquet release in this study were about 10 and 100 times lower than maximum plasma concentrations after systemic intravenous administration of 0.1 mg/kg and 10 mg/kg tiludronate in horses, respectively (*Delguste et al., 2008*).

This study also determined that IVRLP with high or low dose tiludronate did not significantly change synovial fluid cytology variables in comparison to saline control. Changes in synovial fluid cytology variables over time, specifically elevation of total nucleated cell counts 24 h after IVRLP, were observed in all limbs, regardless of specific treatment. These alterations may be due to repeated synoviocenteses or the process of IVRLP itself. Repeated synoviocentesis every 12 (*White et al., 1989*) or 48 h (*Sanchez Teran et al., 2012*) has been shown to significantly increase total nucleated cell counts in comparison to baseline. The elevation in total nucleated cell counts could also be due to diffusion of saline into the synovial cavity following IVRLP. Intraarticular injection of saline has been shown to cause acute synovial inflammation with significant increases in synovial fluid white blood cell counts and total protein concentration 24 h after injection (*Wagner, McIlwraith & Martin, 1982*).

Isotonic saline was chosen as the diluent in this study, as tiludronate may bind calcium if diluted with polyionic, physiologic solutions. A dilution volume of 50 ml was chosen as this is a commonly reported infusion volume for IVRLP with antibiotics (*Levine et al., 2010*; *Kelmer et al., 2013a*; *Kelmer et al., 2013b*). However, a recent study suggested that lower perfusate volumes tend to result in higher synovial fluid concentrations of antibiotics within the perfused area (*Hyde et al., 2013*). Thus, the tiludronate concentrations achieved in this study may not represent what would be achieved if a lower perfusate volume was used.

A limitation of this study was the use of horses that had variable degrees of lameness at a trot prior to any experimental manipulations. Attempts to compensate for this weakness were made by evaluating horses for changes in lameness from baseline. Statistical analysis suggested an increase in lameness score in HDT but not saline control limbs 24 h after treatment when compared to baseline lameness scores and significantly greater lameness in HDT limbs than control limbs at the same time point. However, randomization of treatments resulted in none of the lame limbs being assigned to control treatments and all lame limbs being assigned to tiludronate treatments. Thus, we are unable to determine whether or not IVRLP with tiludronate results in clinically appreciable lameness based on our experiments and further investigation of this aspect is warranted. Nevertheless, results pertaining to synovial fluid tiludronate concentrations and cytology variables are still valid, as lameness by itself is unlikely to influence these variables in non-exercised horses.

Since the therapeutic target tissue of tiludronate is bone, future studies evaluating safety of IVRLP with different doses of tiludronate for bone are essential before determination of possible therapeutic efficacy is undertaken. While the target cells of tiludronate are osteoclasts, there is some evidence that high concentrations of bisphosphonates can induce apoptosis also in osteoblasts (*Patntirapong et al., 2012*), emphasizing the need to ascertain

that bone concentrations of tiludronate after IVRLP are not high enough to negatively impact cells in bone other than osteoclasts.

This study represents a first step to determine the safety of IVRLP with tiludronate for articular cartilage within the perfused area. Findings suggest that IVRLP with either 0.5 mg or 50 mg of tiludronate did not cause synovial inflammation in comparison to saline controls. Further, synovial fluid concentrations of tiludronate after IVRLP with 0.5 mg tiludronate were within a range that can be considered safe for cartilage based on previous *in vitro* data (*Duesterdieck-Zellmer, Driscoll & Ott, 2012*). However, after IVRLP with 50 mg tiludronate, some horses may experience synovial fluid concentrations that may not be safe for articular cartilage of the distal interphalangeal joint.

## ACKNOWLEDGEMENTS

We would like to thank Lauren Hobstetter for her assistance with the experiments.

### Funding

The experiments were funded by the Department of Clinical Sciences of the College of Veterinary Medicine at Oregon State University via 2 clinical resident grants. The funders had no role in study design, data collection and analysis, decision to publish, or preparation of the manuscript.

### Grant Disclosures

The following grant information was disclosed by the authors:
Oregon State University.

### Competing Interests

The authors declare there are no competing interests.

### Author Contributions

- Barbara G. Hunter conceived and designed the experiments, performed the experiments, analyzed the data, contributed reagents/materials/analysis tools, wrote the paper, reviewed drafts of the paper.
- Katja F. Duesterdieck-Zellmer conceived and designed the experiments, performed the experiments, analyzed the data, contributed reagents/materials/analysis tools, wrote the paper, prepared figures and/or tables, reviewed drafts of the paper.
- Maureen K. Larson performed the experiments, analyzed the data, contributed reagents/materials/analysis tools, reviewed drafts of the paper.

### Animal Ethics

The following information was supplied relating to ethical approvals (i.e., approving body and any reference numbers):

Oregon State University's Institutional Animal Care and Use Committee
ACUP# 4280, and ACUP# 4459.

## Supplemental Information

Supplemental information for this article can be found online at http://dx.doi.org/10.7717/peerj.889#supplemental-information.

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
