# Peer review of "Tiludronate concentrations and cytologic findings in synovial fluid after intravenous regional limb perfusion with tiludronate in horses"

_PeerJ, doi:10.7717/peerj.889_

## Round 0.1 · original submission · Minor Revisions

Please consider all the suggestions made by the reviewers in your revised manuscript

Reviewer 1 ·

Basic reporting

This article would probably benefit from a change in tables and figures.
Table 1 is of low interest, because lameness scores for high tiludronate horses from day 2 to 7 are lacking. This table has to be explained fully in the results section.
Figure 1 and 2 titles should mention "normal" reference interval or an equivalent word, and mention where do those intervals come from?
A table may be more appropriate for fig 1 and 3; and a figure with curves would probably be more appropriate for table 2, with more data (more time points).
Lines 230-231: I can't understand the difference between the 2 parts of this sentence, for me, the same thing is repeated twice.

Experimental design

This article is very interesting.
It would have benefit of more time points for collection of synovial fluid after the release of the tourniquet, because it is not possible with the current data to determine the duration of the high tiludronate concentrations in the HDT limbs; and I think that this fact is critical in order to answer the first objective of this study.
Another point is tiludronate analysis. The range for the standard samples is too small (0.5-64ng/mL) and is not correlated with the reported LOQ (10ng/mL). This fact is quite suprising for me.

Validity of the findings

In the summary, it is not correct to state that the "maximal synovial fluid tiludronate concentrations occured 30 minutes post-perfusion". It must be determined with a pharmacokinetic study with more time points.
There is a large disparity between groups, and the lameness issue (tiludronate perfused limbs are lame (line s231-235)) represent a bias.
The MM and results mention synovial fluid sampling at T30min for all joints. For me, it is not possible to simultaneously puncture those 3 joints and the actual times must be specified.

Additional comments

The objectives of your study are relevant and very interesting.
The problem is more data are required. a first possibility is to acquire more time points in order to do a real pharmacokinetic analysis, a second possibility is to perform dosage of cartilage degradation products in order to study more precisely cartilage damage.

Reviewer 2 ·

Basic reporting

Manuscript meets standards although some editing for clarity is required.

Experimental design

Design is an accepted one and fulfills the criteria for experiments: randomization, controls and blinded observers.

Validity of the findings

As for most studies of this nature, the contribution is incremental. Conclusions are stated in an appropriate manner.

Additional comments

It might be worthwhile to indicate the typically used dose administered by regional perfusion in the Introduction - i.e. many use a single 50 mg vial from the packaged 10 vials.
I am not sure why you express the "toxic" dose as 19,000 ng/ml. Why not 19 ug/ml?

·

Basic reporting

No comments

Experimental design

Please could you mention clearly if initial comparability between control and tiludronate treated limbs was systematically assessed ?

Validity of the findings

No comments

Additional comments

This is an interesting and clearly written article.
I only have a few comments and questions :
- A reference to tiludronate plasma values obtained after systemic administration should be added for comparison with concentration data obtained in this study
- A particular comment could be added about inflammation parameters obtained before IVRLP in joints of lame limbs
- Are reference values available for the % of neutrophils in synovial fluid ? (to be mentioned on figures)
- If the HDT group was euthanized, were there no histological examinations on cartilage samples ?
- In the study of Wagner, 1982, did they evaluate the kinetics of the inflammatory response in the joint ? If they did, when occured the inflammatory peak ?
- Line 28 : remove the second « of » ?

---

## Round 0.2 · accepted · Accept

We hope you consider PeerJ for publishing your work in the future.